# Systematic Evaluation of the Distribution of Immune Cells following Subcutaneous Administration of Haemophilus Influenzae Type B Vaccine to Mice

**DOI:** 10.3390/diseases11040139

**Published:** 2023-10-13

**Authors:** Yao He, Yuxiu Zhao, Hongyang Liang, Xue Wang, Haoyue Lan, Dongyang Tian, Yan Li, Hui Wang

**Affiliations:** Beijing Institute of Biological Products Company Limited, Beijing 100176, Chinawx1182716@163.com (X.W.); 18237802315@163.com (D.T.);

**Keywords:** Hib, vaccine, aluminum hydroxide adjuvant, immune response

## Abstract

The Haemophilus influenzae type B (Hib) conjugate vaccine is the most effective way to prevent Hib infection in infants and young children, and it is designed to induce the production of antibodies against polyribosylribitol phosphate (PRP) to protect babies from infection. However, the mechanism of immunity induced by the Hib vaccine is not fully understood. Recently, with the development of the combination diphtheria and tetanus toxoids and acellular pertussis vaccines (DTaP), increasing numbers of manufacturers have begun to develop DTaP-based combination vaccines, like the combination vaccine diphtheria and tetanus toxoids and acellular pertussis and Hib conjugate vaccine (DTaP-Hib), which contains adjuvants. However, the Hib vaccine does not contain adjuvants. It was theorized that the Hib antigen has poor compatibility with aluminum adjuvants for unclear reasons. Therefore, understanding the mechanism of the Hib-vaccine-induced immune response and the influence of adjuvants on the Hib vaccine is of great significance. In this paper, we immunized BalBc mice with either the Hib vaccine or the Hib vaccine that adsorbs aluminum adjuvants (Hib-Al). Here, we analyzed the anti-PRP antibody level and immune response of different cells using cell and cytokine levels. We found that the Hib vaccine could induce a humoral and cellular immune response, and the Hib-Al vaccine could induce greater quantities of IFN-γ, IL-4, and IL-6 and more antigen-specific antibodies through B cells, Th1, Th2, and ILC3s in the spleen. Together, our findings demonstrate the serologic responses and immune response in terms of cell and cytokine levels induced by the Hib vaccine, and they also imply that the addition of aluminum hydroxide adjuvant could enhance the function of the Hib vaccine, which preliminarily reveals the mechanism of immune response induced by the Hib-related vaccine.

## 1. Introduction

Hib is a common parasitic bacterium mainly causing pneumonia and meningitis in children. Early studies documented that children aged 3 months to 5 years are particularly susceptible to Hib meningitis due to their blood not possessing bactericidal activity [1]. The early Hib vaccine composed of the capsular polysaccharide of the organism, PRP, protects older children but does not induce antibodies or provide protection in infants under 18 months. This is because PRP is a T-cell-independent antigen with age-related immunogenicity. To enhance the immunogenicity of PRP, PRP is combined with a carrier (such as DT, TT, or CRM) to transform into a T-cell-dependent antigen, and clinical results have shown that Hib conjugate vaccines are more immunogenic for infants and young children than the unconjugated Hib vaccine [2]. The Hib vaccine can enhance the serum anti-PRP antibody titers through the T-cell-dependent B-cell response. However, whether the other immune cells are induced by the Hib vaccine is not clear. For the past few years, a multi-vaccine approach has become the preferred development direction for vaccines, as the use of combined vaccines could reduce the number of vaccinations required (especially in infants and young children), which is conducive to improving the vaccination rate, and it has great clinical demand and public health value. In recent years, many of these combined vaccines have been based on Hib, including DTaP-Hib, and, in combined vaccines, due to the complex antigenic components used, some components require adjuvants to improve the immunogenicity. Some studies have shown that the aluminum-adjuvant-adsorbed Hib has low immunogenicity compared with the Hib vaccine, and that the immunogenicity of the Hib vaccine is reduced in combined vaccines [3,4]. However, other studies have shown that Hib/DTwP combined vaccines have no significant reduction in anti-PRP antibody levels [5,6]. Therefore, in this study, we aim to determine whether the aluminum adjuvant can influence the immunogenicity of Hib.

The immunogenicity of Hib vaccines is thought to test the anti-PRR antibody level [7,8,9]. Clinically, anti-PRP antibodies in human serum reflect the degree of immunity against Hib infection and have been effectively linked to protection; for example, anti-PRP antibodies at concentrations of ≥0.15 g/mL are thought to provide short-term protection against invasive Hib disease, and the concentration of ≥1.0 g/mL one month after Hib vaccine immunization was considered to confer an immunoprotective effect. Because most studies have focused on serologic responses, the mechanisms underlying the immunogenicity of conjugate vaccines remain poorly understood. It is not clear how the Hib vaccine induces changes at the cell level, like the T-cell memory responses or cytokine production from the immune cells to anti-PRR antibodies, which could provide some information to guide the use of adjuvants to enhance the immunogenicity of the vaccines.

In this work, we immunized BalBc mice with Hib and Hib+Al vaccines and analyzed the different types of antibodies in serum. As the spleen is the largest secondary lymphoid organ with a high density of antigen-presenting cells and lymphocytes, and it is responsible for blood filtration and producing an immune response to pathogens [10], we also analyzed the immune response in the spleen. The humoral and cellular immune response to the Hib and Hib+Al vaccines could address the question of whether the addition of an aluminum adjuvant to the Hib vaccine could have a protective function. This study provides theoretical support for the study and development of combined vaccines.

## 2. Results

### 2.1. Hib and Aluminum-Adjuvant-Adsorbed Hib Vaccines Both Induce Humoral Immune Response

B lymphocytes need to undergo a germinal central response to have the ability to produce high-affinity antibodies specific to pathogens, thereby protecting the body from pathogen infection. To test whether aluminum hydroxide Al(OH)_3_ affects Hib immunogenicity, an Al(OH)_3_-adsorbed Hib vaccine (Hib+Al) and a Hib vaccine were administered to mice, and the germinal center B cells (GCBs) and the anti-PRP antibody level were analyzed. The results show that both the Hib and Hib+Al vaccines induce a humoral immune response, including anti-PRP antibody and GCBs. However, the Hib+Al vaccine induced a higher level of anti-PRP antibody (Figure 1A,B) and more GCB cells (Figure 1C,D) than the Hib vaccine, suggesting that the aluminum hydroxide adjuvant could enhance the humoral immune response in the Hib vaccine. Different antibodies were induced after the antigen stimulation. Here, we analyzed the antibodies of different isotypes, including IgM, IgG1, IgG2a, and IgG2b. The results show that IgM caused no difference in the Hib and Hib+Al groups (Figure 2A), and the aluminum adjuvant improved the expression of IgG1, IgG2a, and IgG2b (Figure 2B–D), which was consistent with the above results (Figure 1B). Together, these data confirm that the addition of aluminum adjuvant could improve the humoral immune response in mice.

### 2.2. The Effect of Aluminum Adjuvant on Hib-Induced Cytokine Expression

IgG2b was mainly induced by Th1 cells, which could produce IL-2 and IFN-γ, while IgG1 and IgG2a were mainly elicited by Th2 cells, which could produce IL-4. The ratio of IgG1/IgG2a could be used to evaluate whether Th1 or Th2 was biased in the vaccine-induced immune response. IgG2a was upregulated in the Hib-Al vaccine groups, which indicated that the aluminum hydroxide adjuvant may induce a Th1 response. To further explore the effect of the aluminum adjuvant on the Hib vaccine, the immune cells in the spleens of the immunized mice were stimulated with Hib. The cytokine expression levels were analyzed. Through ELISpot assay, we found that the Hib vaccine could induce IL-4 expression, which was consistent with the high expression of the antibody in Hib-immunized mice (Figure 3C,D). Also, the Hib vaccine showed an upward trend for inducing IFN-γ expression, although it showed no significant difference compared to the control (Figure 3A,B). However, the Hib+Al vaccine could induce both IL-4 and IFN-γ expression (Figure 3B,D), suggesting that the aluminum adjuvant could enhance both the humoral and cellular immune response. Furthermore, the immune cells in the spleen were stimulated with Hib for 24 h, and we tested different cytokine levels using a CBA kit (Figure 4). The results show that the Hib and Hib+Al vaccines induced IL-2, IL-6, and IFN-γ expression(Figure 4B,D,E), suggesting that the Hib vaccine could induce cellular immune response and antibody production. In addition, the Hib+Al vaccine could also induce IL-4 expression(Figure 4C), and it produced more IL-2 than the Hib vaccine. These results indicate that the aluminum adjuvant could induce a stronger immune response, including the cellular and humoral immune response.

To determine whether the aluminum adjuvant could disrupt the immune system, we tested the Th1/Th2/Th17 cytokine levels in serum from the immunized mice. The results showed that there were no significant differences in Hib+Al- and Hib-vaccine-immunized mice (Figure 1), indicating that the aluminum adjuvant in the Hib vaccine could also maintain the homeostasis of immune cells. Together, these results demonstrate that the aluminum adjuvant could improve Th1 and Th2 cytokine expression.

### 2.3. The Distribution of Immune Cells following Administration of Hib and Hib+Al Vaccine

The Hib vaccine could enhance the anti-PRP antibody titers in serum. However, the distributions of other immune cells induced by the Hib vaccine were unclear. Our results above have proven that the addition of aluminum adjuvant was beneficial to the production of antibodies and the cellular immune response. Next, we explored which types of immune cells were changed after the mice were immunized with the Hib or Hib+Al vaccines(Figure 5). We found that the Hib vaccine could increase CD4+ T cells, neutrophil, and ILC2 cells and decrease ILC1 cells(Figure 5A,C,E,G). We also found that the Hib vaccine could decrease the MHC II expression in the CD11b-CD45+ population and increased the MHC II expression in the CD11b+CD45+ population, which suggests that the Hib vaccine could enhance the MHC II expression in myeloid cells (Figure 2). CD4+ T cells were important both in humoral and cellular immune responses. The neutrophils played an important role in the innate immune responses, and it has been reported that they are required during vaccination for host protection and for promoting the antibody responses at immunization sites [11]. These results indicated that the Hib vaccines could induce a high level of antibodies. However, there were no differences in CD8+T, DC, monocytes, or ILC3s between the Hib and control groups (Figure 5A,B,D,E,F). In comparison with the Hib vaccine, the Hib+Al vaccine significantly decreased the percentage of ILC3s. It has been reported that ILC3 could influence the Tfh function and inhibit the antibody production of GCB cells through PDL1 and MHCII, suggesting that the aluminum adjuvant may also induce the antibody through inhibition of ILC3s. These results show that the Hib vaccines could induce the innate and adaptive immune responses, and the aluminum adjuvant could promote the immune response by regulating the ILC3s.

## 3. Discussion

Hib is one of the common causes of pneumonia, meningitis, and other serious infections in children, and vaccination is one of the most effective means of preventing Hib disease worldwide [12]. With the development of DTaP-series combination vaccines [13], understanding the mechanisms of the Hib vaccine and the interaction of the adjuvant with Hib is of great significance to the development of combination vaccines. In this work, we analyzed the immune response induced by the Hib vaccine and found that the Hib vaccine induced the formation of GCBs, which are a core part of humoral immunity, and their differentiation to plasma cells and memory B cells determines the quantity and quality of final antibodies. Consistent with the formation of GCBs, the Hib vaccine also promotes the production of anti-PRP antibodies, including IgG1, IgG2a, IgG2b, and IgM. With the exception of humoral immunity, it can also improve cellular immunity. After immunization with the Hib vaccine, we evaluated the Th1 response (IL-2, IFN-γ). IL-2 promoted the differentiation of CD4+ T cells into Th1 and Th2 cells, promoted CD+4 and CD8+ T cell activation and memory phenotype formation, and could also inhibit the differentiation of CD4+ T cells into inflammatory Th17 cells [14]. Therefore, the cytokines could improve the function of the Hib vaccine. These results show that the Hib vaccine protects the body both through humoral and cellular immunity.

With the development of DTaP/Hib combination vaccines, a number of noteworthy problems have arisen. There are many components of the combined vaccine, and these different components have numerous properties. Additionally, the DTaP antigens require an adjuvant to activate the immune system, but the Hib vaccine does not contain adjuvants. Many reports have shown that the functions of Hib are decreased in DTaP/Hib combined vaccines [15]. For example, in trials with the Hib/DTaP3 combined vaccine, the results showed significant reduction in Hib antibodies, especially in pre-term infants, compared with the individual injection of Hib and DTaP3 vaccines. Also, some clinical studies show that there were no significant reductions in anti-PRP antibody levels using the Hib/DTwP combined vaccines [5,6,16]. The problem of reduced Hib antibodies in combination vaccines is believed to be due to interference between antigens or pre-existing immune responses to carrier proteins (conjugated to Hib antigens, such as TT) significantly reducing the production of anti-PRP antibodies in some studies. However, reducing the dose of TT did not influence the immunogenicity of Hib either [17], while some issues are believed to be caused by the incompatibility of Hib and adjuvants [18,19]. Therefore, whether adjuvants can influence the Hib vaccine is a question worth studying. Here, we used the Hib+Al vaccine to immunize mice. The efficacy of the vaccine was evaluated from different dimensions. The results show that the addition of the adjuvant promoted greater GCB and anti-PRP antibodies compared with the Hib vaccine. The different subtypes of IgG (IgG1, IgG2a, IgG2b) also demonstrated a rising trend in the Hib+Al group. The IgG1 was associated with a Th2-type immune response, and IgG2a was stimulated during Th1-type immune responses [20,21,22]. The Th1 subset could produce IFN-γ and IL-2 [23]. Therefore, the aluminum adjuvant may induce the Th1 and Th2 responses, which was proved by the increase in IL-2 and IFN-γ expression in the Hib+Al group. The aluminum adjuvant played an effective role in the Hib vaccine, which is inconsistent with previous reports [3]. Hib conjugate vaccines have different immunological profiles depending on the manufacturer, and the immunogenicity mainly depends on the carrier protein type, binding method, conjugate molecular size, and glycoprotein ratio, among which the conjugate molecular size is crucial for the immunogenicity of the vaccine. Therefore, the different effects of aluminum adjuvants on Hib vaccines may be influenced by animal models, aluminum adjuvant types, immune programs, and the antigenic components. The mechanisms of the effects of aluminum adjuvants on Hib vaccines by different manufacturers need to be further explored. The Hib vaccine demonstrates good safety and efficacy. However, it is still unclear which cells are induced by the Hib vaccine. In this work, we also found that CD4+ T cells, neutrophils, and ILC2 were upregulated after immunization with the Hib vaccine. CD4+ T cells can promote the antibody production of B cells, and they can also differentiate into Th17 or Treg cells, inhibiting or promoting inflammation. We also measured the IL-10 and IL-17 secretion in the serum and in the supernatants of cell secretions. There were no significant differences between the Hib and control groups, which suggests that the Hib vaccine may not induce high levels of Th17 or Treg cells. However, this point needs to be examined further. The neutrophil also plays an important role in the antibody response [24]. The addition of the aluminum adjuvant reduced the ILC3 levels in the spleen. It has been reported that ICL3 expresses the major histocompatibility complex class II and PD-1, which regulates the CD4+ T cell responses [25,26] and could inhibit the function of Tfh and then influence GCB formation [27]. Therefore, in this study, the aluminum adjuvant may have influenced ILC3 and promoted the production of antibodies via B cells.

In summary, our data illustrate the mechanism of Hib vaccine induction and that it can induce both the Th1 and Th2 cell responses while also promoting the neutrophil and ILC2 response. The addition of the aluminum adjuvant upregulated both the Th1 and Th2 cell response and decreased the ILC3 expression. These results reveal that the addition of the adjuvant in the Hib vaccine was beneficial for inducing a stronger immune response, which provides theoretical support for the development of subsequent combination vaccines. The interaction of Hib and DTaP components in the immune response should be further explored in order to develop better Hib-containing combination vaccines.

## 4. Materials and Methods

### 4.1. Vaccines

A Hib vaccine containing 10 μg of PRP/0.5 mL conjugated to TT, with a protein/polysaccharide ratio of 2–3, was used. The Hib vaccine was absorbed with aluminum hydroxide adjuvant.

### 4.2. Animals and Immunization

Mice were purchased from Beijing Weitonglihua Experimental Animal Technology Co., Ltd., Beijing, China and then maintained in a specific-pathogen-free (SPF) environment at the Laboratory Animal Center of Beijing Institute of Biological Products Co., Ltd., Beijing, China. All of the mice were female, Balb/C, and they weighed 12~14 g. The mice were immunized with vaccines (2.5 μg/dose) on day 0 and day 14, and the sera were extracted at day 21. The total number of animals per group is presented in Table 1.

### 4.3. Reagents

The reagent information is presented in Table 2.

### 4.4. ELISA

Antibodies to PRP were detected by ELISA. Briefly, plates were coated with PRP overnight at 4 °C (0.5 Lf/mL). After blocking for 1 h with PBS containing 1% BSA, the gradient-diluted serum samples were added to the plate, and, after 1 h, anti-mouse HRP-IgG (1:1000), HRP-IgG1 (1:5000), HRP-IgG2a (1:5000), HRP-IgG2b (1:5000), and HRP-IgM (1:5000) were added to the plates and incubated for 30 min, RT. After washing, TME and stop solution were added to the plates. The absorbance was measured at 450 nm and OD630.

### 4.5. FACS

The spleens were analyzed on day 28 after initial immunity. The spleen cells were lysed using RBC (Biolegend, San Diego, CA, USA, 420301), and then the cells were blocked with blocking antibody (CD16/32) for 30 min at 4–8 °C. After washing, the cells were stained with Fixable Viability Dye eFluor 506, which was added to exclude the dead cells, and then the cells were stained with other antibodies using the appropriate antibody on ice for 30 min. Flow cytometry was performed on a CytoFLEX S instrument (Beckman, Brea, CA, USA) and analyzed with CytoFLEX S FlowJo software (1.0). The detailed protocol was described by [28].

### 4.6. ELISpot

Immune cells in the spleen were seeded into the 96-well plates (5 × 10^5^ cells per well) and incubated with PRP solution (4 mg/mL) in DMEM medium (Gibco, Thermo Fisher Scientific, Waltham, MA, USA, C11995500BT) containing 1% penicillin–streptomycin (Procell, PB180120) and 10% FBS (Gibco, Thermo Fisher Scientific, Waltham, MA, USA). After 36 h of stimulation, the IFN-γ and IL-4 produced by immune cells were tested with the kits (Mouse IFN-γ precoated ELISpot kit and the Mouse IL-4 precoated ELISpot kit) (Dakewe Group, Shenzhen, China).

### 4.7. CBA Analysis

Immune cells in the spleen were seeded into the 96-well plates (5 × 10^5^ cells per well) and incubated with PRP solution (4 mg/mL) in DMEM medium (Gibco, Thermo Fisher Scientific, Waltham, MA, USA, C11995500BT) containing 1% penicillin–streptomycin (Procell, PB180120) and 10% FBS (Gibco, Thermo Fisher Scientific, Waltham, MA, USA). After 36 h of stimulation, the cytokines in the supernatant were tested with a CBA kit (BD biosciences, USA). Briefly, we mixed the capture beads and added 50 µL of the sample and detection reagent to the tubes followed by incubation for 2 h at room temperature. The samples were then tested using a CytoFLEX S instrument (Beckman, Brea, CA, USA).

### 4.8. Statistics

Statistical analysis was performed using GraphPad Prism 8. A two-tailed Mann–Whitney test was used to determine significance. Error bars represent SEM. * *p* < 0.05; ** *p* < 0.01; *** *p* < 0.001.

## 5. Conclusions

We systematically evaluated the distribution of immune cells after subcutaneous injection of Hib vaccine, and found that the Hib vaccine could induce both the Th1 and Th2 cell responses, also the neutrophil and ILC2 response. The addition of the aluminum adjuvant could promote the immune response of Hib vaccine which was beneficial for inducing a stronger immune response.

## Data Availability

The data presented in this study are available within the article. Limitations of the Study: Our study only evaluated the Hib-vaccine-induced humoral and cellular immunity in mice.

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
