# Peer review of "Systematic Evaluation of the Distribution of Immune Cells following Subcutaneous Administration of Haemophilus Influenzae Type B Vaccine to Mice"

_diseases, 2023, doi:10.3390/diseases11040139_

Round 1

Reviewer 1 Report

Authors of the presented manuscript evaluated the humoral and cellular immune responses of the sole Haemophilus influenzae type B (Hib) vaccine or in conjugation with aluminum as adjuvant. The manuscript is well-written and considered valuable within its field. Few suggestions are presented:

1.      Authors are advised to explore CD11b+/Gr1+ population in terms of mature and immature monocytes/macrophages based on differential Ly6C and Ly6G levels, rather than those of Gr1 surface marker. This would further highlight the vaccine impact on inflammatory-cellular immune responses.

2.      In terms of CD11b/CD45 population, narration on the levels and homogeneity of MHCII across 4 different population combinations (CD11b+/-CD45+/-) would also be beneficial to explore mechanistic role of Hib across the myeloid lineage.

3.      Authors are advised to present a rational for investigating immune cellular expressions being isolated from the spleen and not from bone marrow tissues.

Minor editing of English language required

Reviewer 2 Report

Systematically Evaluate the Distribution of Immune Cells Following Subcutaneous Administration of Haemophilus influenzae type B Vaccine to Mice

Comments:

Title can be improved grammatically to “Systematic Evaluation of the Distribution of Immune Cells Following Subcutaneous Administration of Haemophilus influenzae type B Vaccine to Mice”.

Material and methods look good but, in statistical analysis it can be pointed out that number of samples were less than optimum to apply tests.

In “Introduction” or “Discussion” author could explain contradictory report about Aluminium adjuvants and Combined vaccine with respect to immunity, text seems ambiguous, Authors could explain resemblance with available literature and rational for contradiction.

Under heading “Introduction”, on page 2, line 57, sentence “Evaluate the immunogenicity of Hib vaccines is thought to test the anti-PRR antibody Level" e the immunogenicity of Hib vaccines is thought to test the anti-PRR antibody Level" can be grammatically improved with respect to the context.

As mentioned in comment section

Author Response

Rebuttal lette

Tanks for your suggestion, and we have added the deficiencies in the paper. The specific response is as follows:

  1. Title can be improved grammatically to “Systematic Evaluation of the Distribution of Immune Cells Following Subcutaneous Administration of Haemophilus influenzae type B Vaccine to Mice”.

Reply: Thanks for your suggestion. We have improved the grammar in the title.

  1. Material and methods look good but, in statistical analysis it can be pointed out that number of samples were less than optimum to apply tests.

Reply: Thanks for your suggestion. We have added the total number of animals per group in the method,as shown in the table 1. And except the GCB analysis experiment, we use the 4 mice, other experiments used

Table 1. Total number of animals per group.

Experiment

Group

Total Number

Tissue

Immunization experiments

Hib+Al

10

/

Hib

10

Ctrl

10

Total anti-PRP IgG analysis

Hib+Al

10

Serum

Hib

10

Antibody analysis of different subtypes

Hib+Al

5

Serum

Hib

5

CBA

Hib+Al

5

Serum

Hib

5

Ctrl

5

ELISPOT

Hib+Al

5

Spleen

Hib

5

Ctrl

5

FACS

Other immune cells analysis

5

Spleen

GCB analysis

4

Fig 1. Hib and Hib+Al vaccine-induced GCB expression

The representative flow cytometry chart and the percentage of GCB were shown, Error bars represent the SEM. ∗p < 0.05; ∗∗p < 0.01; ∗∗∗p < 0.001 (Student’s t test); ns, no significant difference.

  1. In “Introduction” or “Discussion” author could explain contradictory report about Aluminium adjuvants and Combined vaccine with respect to immunity, text seems ambiguous, Authors could explain resemblance with available literature and rational for contradiction.

Reply: Thanks for your suggestion. And we added the analysis of the Hib vaccine induced immune response  Aluminium adjuvants and Combined vaccine with respect to immunity on page 8, line 213. And the analysis is as follows:

Many reports have shown that the function of Hib were decreased in DTaP/Hib combined vaccine, like the trials with the Hib/DTaP3 combined vaccine, the results showed significant reduction in Hib antibody, especially in pre-term infants, compared with the inject individually of Hib and DTaP3 vaccines. Also, some clinical studies show that there were no significant reduction in anti-PRP antibody levels using the Hib/DTwP combined vaccines. The problem of reduced Hib antibodies in combination vaccines is believed to be interference between antigens or pre-existing immune responses to carrier proteins (conjugated to Hib antigens, such as TT) signifi-cantly reduced the production of anti-PRP antibodies in some studies. However, reducing the dose of TT also did not influence the immunogenicity for Hib. And some are believed to be caused by the incompatibility of Hib and adjuvants. Hib conjugate vaccines have different immunological profiles depending on the manufacturer, and the immunogenicity mainly depends on the carrier protein type, binding method, conjugate molecular size and glycoprotein ratio, among which the conjugate molecular size is crucial for the immunogenicity of the vaccine. So the different effects of aluminum adjuvants on Hib vaccines may be influenced by animal models, aluminum adjuvant types, immune programs, and also the antigenic components. The mechanisms of effects of aluminum adjuvants on Hib vaccines by different manufacturer need to be further explored.

  1. Under heading “Introduction”, on page 2, line 57, sentence “Evaluate the immunogenicity of Hib vaccines is thought to test the anti-PRR antibody Level" e the immunogenicity of Hib vaccines is thought to test the anti-PRR antibody Level" can be grammatically improved with respect to the context.

Reply: Thanks for your suggestion. We have improved the grammar, and could see line 57, on page 2 of the article for details.

Reviewer 3 Report

The manuscript has been well written and documented. The results sustain the conclusion. The manuscript is basically acceptable as it is.

Ok

Author Response

Thanks for your suggestion

Round 2

Reviewer 1 Report

Authors adequately responded to comments and suggestions